# NGS-Based Molecular Karyotyping of Multiple Myeloma: Results from the GEM12 Clinical Trial

**DOI:** 10.3390/cancers14205169

**Published:** 2022-10-21

**Authors:** Juan Manuel Rosa-Rosa, Isabel Cuenca, Alejandro Medina, Iria Vázquez, Andrea Sánchez-delaCruz, Natalia Buenache, Ricardo Sánchez, Cristina Jiménez, Laura Rosiñol, Norma C. Gutiérrez, Yanira Ruiz-Heredia, Santiago Barrio, Albert Oriol, Maria-Luisa Martin-Ramos, María-Jesús Blanchard, Rosa Ayala, Rafael Ríos-Tamayo, Anna Sureda, Miguel-Teodoro Hernández, Javier de la Rubia, Gorka Alkorta-Aranburu, Xabier Agirre, Joan Bladé, María-Victoria Mateos, Juan-José Lahuerta, Jesús F. San-Miguel, María-José Calasanz, Ramón Garcia-Sanz, Joaquín Martínez-Lopez

**Affiliations:** 1Hematology Department, Hospital 12 de Octubre, 28041 Madrid, Spain; 2H12O–CNIO Hematological Malignancies Clinical Research Unit, Spanish National Cancer Research (CNIO), 29029 Madrid, Spain; 3Unidad de Biología Molecular-HLA, Laboratorio de Hematología, Hospital Universitario de Salamanca, 37007 Salamanca, Spain; 4Center for Applied Medical Research (CIMA) LAB Diagnostics, Universidad de Navarra, 31008 Pamplona, Spain; 5Cancer Research Institute of Salamanca-IBMCC (USAL-CSIC), 37007 Salamanca, Spain; 6Centro de Investigación Biomédica en Red Cáncer (CIBERONC), 28029 Madrid, Spain; 7Hospital Clinic de Barcelona, 08036 Barcelona, Spain; 8August Pi i Sunyer Biomedical Research Institute (IDIBAPS), 08036 Barcelona, Spain; 9Clinical Research Support Unit, Institut Català d’Oncologia, 08036 Barcelona, Spain; 10Servicio de Genética, Hospital 12 de Octubre, 28041 Madrid, Spain; 11Hematology Department, Hospital Ramón y Cajal, 28034 Madrid, Spain; 12Hospital Universitario Puerta de Hierro, 28222 Majadahonda, Spain; 13Institut Catalá d’Oncologia-l’Hospitalet, Institut de Investigació Biomèdica de Bellvitge (IDIBELL), Universitat de Barcelona, 08908 Barcelona, Spain; 14Hospital Universitario de Canarias, 38320 Santa Cruz de Tenerife, Spain; 15Hospital La Fe, 46026 Valencia, Spain; 16Instituto de Investigaciones Sanitarias de Navarra (IdiSNA), Universidad de Navarra, 31008 Pamplona, Spain

**Keywords:** multiple myeloma, next generation sequencing, cytogenetics, high-risk

## Abstract

**Simple Summary:**

Multiple Myeloma (MM) is considered an incurable chronic disease, which prognosis depends on the presence of different genomic alterations. To accomplish a complete molecular diagnosis in a single essay, we have designed and validated a capture-based NGS approach to reliably identify pathogenic mutations (SNVs and indels), genomic alterations (CNVs and chromosomic translocations), and *IGH* rearrangements. We have observed a good correlation of the results obtained using our capture panel with data obtained by both FISH and WES techniques. In this study, the molecular classification performed using our approach was significantly associated with the stratification and outcome of MM patients. Additionally, this panel has been proven to detect specific *IGH* rearrangements that could be used as biomarkers in patient follow-ups through minimal residual disease (MRD) assays. In conclusion, we think that MM patients could benefit from the use of this capture-based NGS approach with a more accurate, single-essay molecular diagnosis.

**Abstract:**

Next-generation sequencing (NGS) has greatly improved our ability to detect the genomic aberrations occurring in multiple myeloma (MM); however, its transfer to routine clinical labs and its validation in clinical trials remains to be established. We designed a capture-based NGS targeted panel to identify, in a single assay, known genetic alterations for the prognostic stratification of MM. The NGS panel was designed for the simultaneous study of single nucleotide and copy number variations, insertions and deletions, chromosomal translocations and V(D)J rearrangements. The panel was validated using a cohort of 149 MM patients enrolled in the GEM2012MENOS65 clinical trial. The results showed great global accuracy, with positive and negative predictive values close to 90% when compared with available data from fluorescence in situ hybridization and whole-exome sequencing. While the treatments used in the clinical trial showed high efficacy, patients defined as high-risk by the panel had shorter progression-free survival (*p* = 0.0015). As expected, the mutational status of *TP53* was significant in predicting patient outcomes (*p* = 0.021). The NGS panel also efficiently detected clonal *IGH* rearrangements in 81% of patients. In conclusion, molecular karyotyping using a targeted NGS panel can identify relevant prognostic chromosomal abnormalities and translocations for the clinical management of MM patients.

## 1. Introduction

Multiple myeloma (MM) is a heterogeneous disease with a complex clonal and subclonal architecture and with few recurrent mutations [1,2]. The expansion of genetic events, including translocations, copy number variations (CNVs) and point mutations, facilitates tumor progression and is also responsible for the short remissions and numerous relapses along the disease timeline [3]. Some of these genetic abnormalities have a major impact on prognosis [4].

However, only a few translocations and CNVs define outcome risk in patients with MM. Cytogenetic aberrations are routinely evaluated using fluorescence in situ hybridization (FISH) panels and single-nucleotide polymorphism (SNP) microarrays in bone marrow (BM)-isolated plasma cells of MM patients [4,5]. Deletion of 17p (del(17p)) and translocations t(4;14), t(4;16) and t(14;20) were identified as poor prognostic biomarkers; likewise, gain(1q) and del(1p) are considered high-risk features for some groups [3]. The coexistence of different chromosomal aberrations can also modify the prognostic risk––for example, trisomy of chromosome 3 or 5 is associated with a better prognosis in patients with tumors displaying other high-risk genetic features [6,7], whereas del(1p32) in patients with t(4;14), as well as del(6q) in patients with del(17p) and t(11;14) [8,9], increases the risk. In this context, the development of new molecular technologies has facilitated the classification of very high-risk patients who are characterized by either biallelic inactivation of *TP53* or amplification (≥4 copies) of 1q21 in patients presenting ISS-III (International Stage System) [10]. The presence of this “double-hit” high-risk signature is currently considered a key prognostic factor [10,11,12].

The advent of next-generation sequencing (NGS) has ushered in a new era of diagnostic understanding and classification of the genomic landscape of MM [3] and has revealed the presence of unique and shared mutations in coexisting subclones of malignant/pathogenic plasma cells (PPCs) and the absence of a universal driver mutation [2,3,13,14]. While several mutated genes have been described to date in MM, only loss-of-function mutations of *TP53* have had a clear prognostic impact. Accordingly, obtaining a complete genomic profile of MM, including single nucleotide variations (SNVs), insertion/deletion mutations (Indels), CNVs and translocations, would be crucial for the accurate diagnostic and prognostic stratification of MM patients. Along this line, there is great interest in the development of an NGS panel to identify––in a single assay––all possible chromosomal alterations (so-called molecular karyotype), which would considerably simplify the genomic diagnosis of patients with MM in a cost-effective manner [15,16,17] and would likely impact future targeted therapy approaches [18]. Although some efforts were made in this direction [15,16,17], the results have not been validated in cohorts of homogeneously treated patients with MM from clinical trials.

In the present study, we designed a capture-based targeted NGS panel to identify relevant genetic aberrations for the prognostic stratification of MM in a single assay. The panel aims to simultaneously detect SNVs, indels, CNVs, chromosomal translocations and V(D)J clonotypes [19] with a higher resolution and a smaller amount of required starting material than conventional techniques. NGS molecular karyotyping was validated and compared with data from FISH and whole-exome sequencing (WES). We also clinically validated the MM-specific NGS panel in samples obtained at diagnosis from patients with MM enrolled in the GEM12 clinical trial [20]. Our findings suggest that this new approach could eventually replace the conventional and standard procedures used in laboratories for the molecular diagnosis of MM.

## 2. Materials and Methods

### 2.1. Patient Cohort and Samples

We studied genomic (g)DNA from CD138^+^ plasma cells of BM aspirates from 149 patients with newly diagnosed MM enrolled in the GEM2012MENOS65 clinical trial [21]. Patients with active newly diagnosed MM were treated according to the PETHEMA GEM2012MENOS65 and the companion PETHEMA/GEM2014MAIN clinical trials [21]. Each study site’s independent ethics committee reviewed and approved the protocol, amendments, and informed consent forms. The study was designed and conducted per the ethical principles of the Declaration of Helsinki and the International Council for Harmonization Guidelines. The study was registered at www.clinicaltrials.gov (accessed on 27 September 2017) as #NCT01916252 and EudraCT as #2012-005683-10. To summarize, patients received VRd plus autologous stem cell transplantation, VRd consolidation and maintenance with Lenalidomide or Lenalidomide plus Ixazomib.

The clinicopathological features of the patients are summarized in Table 1. The average age at diagnosis was 60 years (range: 42–65), with a similar frequency for males (53%) and females (47%). The clinical characteristics of the patient cohort were representative of the patients included in the clinical trial [21]. Plasma cells from BM aspirates were enriched using anti-CD138^+^ immunomagnetic beads, obtaining a purity >85% in all cases. DNA was extracted using the AllPrep DNA/RNA Mini Kit (Qiagen, Hilden, Germany) and capture libraries were generated with the SureSelectQ^XT^ Reagent Kit (Agilent Technologies, Santa Clara, CA, USA) using 50 ng of genomic DNA. Libraries were multiplexed and sequenced on an Illumina NextSeq 500 platform (High Output Kit v2, Illumina, San Diego, CA, USA) in 150 bp paired-end mode.

### 2.2. Capture Next-Generation Sequencing Panel Design

Two panel versions were used in this study; however, differences between panel versions are mainly focused on the improvement of coverage in the 14q32 region for translocation identification. Design was performed to detect the most common and relevant aberrations known to occur in MM [3], covering the coding regions of 26 genes involved in the development and progression of tumors (*ATM, ATR, ATRIP, BCL7A, BRAF, CCND1, CYLD, DIS3, EGR1, FAM46C, FGFR3, HIST1H1E, IRF4, KRAS, LTB, MAX, NRAS, NRM, PRDM1, PRKD2, RB1, TRAF3, ZFHX4, CRBN* and *NFKB2*). Additionally, the complete exonic region was covered for *TP53*. Genes considered to be treatment targets or candidates for drug resistance in MM (*PSMD1, XBP1, PSMB5, PSMC2, PSMC6, DDB1, IKZF1* and *IKZF3*) [16,17,18] and genes related to new immunotherapy treatments (*CD38*, *CD19* and *SLAMF7*) were also included [22]. In addition, specific regions within canonical *IGH* locus (14q32) were included to capture *IGH* rearrangements, as well as germline polymorphisms distributed throughout the genome, to have a good representation of each chromosome to reliably detect CNVs. The total target size of the final design was 400 kb (SureDesign, https://earray.chem.agilent.com/suredesign/index.htm (accessed on 12 February 2019). Design and order information are available upon request to corresponding authors.

### 2.3. Bioinformatic Analyses

Raw FASTQ files were evaluated using quality control checks from FastQC [23] and Trimmomatic [24] was used to remove low-quality bases, adapters and other technical sequencing artifacts. Each FASTQ file was aligned to the human reference genome (GRCh38/hg38) using Novoalign (http://www.novocraft.com/products/novoalign/ (accessed on 1 March 2020). Optical and PCR duplication elimination was carried out with Sambamba [25]. SNVs and indels were identified through the combination of VarScan2 [26] and bcftools [27]. Variants were annotated using several functional (RefSeq, Pfam), population-related (dbSNP, 1000 Genomes, ESP6500, ExAC, genomeAD), in silico functional impact prediction (dbNSFP, dbscSNV), and cancer-related (COSMIC and ICGC) databases. Only variants described in the Catalog Of Somatic Mutations In Cancer database with prediction of the effect on protein function (codon stop, splicing, frameshift, nonsynonymous and Indels) were selected and filtered considering the following parameters: *GMAF* population frequency ≤0.01 (obtained from any database), total coverage at position ≥50, frequency of the variant in the sample ≥0.01, absence of annotations indicative of an artifactual-type distribution/quality of reads, and distance to the design bed regions ≤200. Potential CNVs were analyzed through an adaptation of the CONTRA v2.0.8 algorithm [28], using all samples as baseline. Breakpoint analysis was carried out with LUMPY [29] and Delly [30] to detect other potential rearrangements, selecting those events supported by more than 10 reads, followed by inspection of other rearrangement values (homology of consensus sequence, balance in the number of reads supporting each side of the breakpoint, frequency of the breakpoint in the cohort, etc.). *IGH* rearrangements were evaluated using MiXCR [31] and Vidjil [31], ruling out those supported by <5 reads or presenting with a frequency <0.01.

### 2.4. Fluorescence In Situ Hybridization

The FISH panels for MM included probes to detect different translocations (t(4;14), t(6;14), t(11;14), t(14;16), t(14;20)) as well as alterations in chromosomal regions, such as chr1 (*CDKN2C*/*CKS1B*), chr8 (*MYC* BA), chr11 (*CCND1*), chr13 (*RB1*/*DLEU*/*LAMP*), chr14 (IGH BA), and chr17 (Iso(17p)), from MetaSystems Probes (Altlussheim, Germany). Only samples with clonality >15% were considered positive, with the exception of *TP53* (clonality >20%), as specified in the guidelines from the Spanish Myeloma Research Group (GEM) [32,33] to perform a better comparison against the whole clinical cohort.

### 2.5. Whole-Exome Sequencing

WES data were obtained from peripheral blood cells (control) and PPC (tumor) samples from 51 patients to validate the mutations, CNVs and translocations. Exome sequencing was performed using the SureSelect Human All Exon Kit (v5, 51 Mb, Agilent Technologies, Santa Clara, CA, USA) by Macrogen Inc. (Geumcheon-gu, Seoul, Korea), with an output of 0.6 Gb per sample. Bioinformatics analyses were performed as described above, except for total coverage at position that we used ≥10 instead of ≥50. Those variants observed in control exome data were ruled out for consideration as germline variants. Loss of heterozygosity analysis was performed based on a selection of germline SNPs across the genome (minor allele frequency >0.3 and <0.7). Only CNVs observed on chromosomes 1 and 17 were reported.

### 2.6. Statistical Analyses

Statistical analyses were performed using RStudio and specific libraries from Bioconductor (https://bioconductor.org/ (accessed on 25 May 2021). Progression-free survival (PFS) was evaluated using Kaplan–Meier curves and two-sided log-rank tests. Multivariate analysis to determine hazard ratios was performed using the *coxph* and *ggforest* functions in R. The results were considered statistically significant at *p* < 0.05. Cytogenetic high-risk profile was considered if the selected CD138^+^ cells presented a translocation involving either chr4 or chr16 with chr14 (t(4;14)(p16;q32) and t(14;16)(q32;q23) or the presence of del(17p).

## 3. Results

### 3.1. Targeted Capture Next-Generation Sequencing Panel and Exome Statistics

Sequencing statistics were calculated for each panel version (see Appendix A). Panel 1 had an average of 7.3 million reads (range 3–14 million), which translated into a mean base depth of 1746 (range 567–3272) with an average fraction of bases of 88.5% (range 84–94%) at >200×. The average reads in Panel 2 were 33 million for PPCs, resulting in a mean base depth of 2427× (range 66–5967×) and 71% (range 8.58–88.85%) of bases at >200×. Although the sequencing results were notably impacted by the quality of the sample DNA, the mean base depth was >500× for 91% (136 out of 149) of PPCs and the fraction of bases at >200× was >75% in 88% (131 out of 149) of cases from both panel versions. In addition, we estimated the mean base depth per million reads to be 239 (range 98–300) and 75 (range 2–165) for panel versions 1 and 2, respectively. Regarding the *IGH* genomic region, we observed an average of 1723 (range 709–3451) and 8691 (range 142–20,634) mapped reads in panels 1 and 2, respectively (see Appendix A). Exome statistics are shown in Appendix A. No significant differences were found between tumor and control sequencing data regarding the number of reads, coverage or depth distribution. The mean base depth was 66× (range 63–116×) and 80× (range 60–115×) for tumors and controls, respectively, indicating that no bias would be expected in SNV or CNV analyses.

### 3.2. Molecular Karyotyping Characterization of Patients with MM Using NGS Panels

The global molecular characterization of the 130 patients with at least one molecular alteration is shown in Figure 1. Below, we describe the different types of alterations, including SNVs, indels, CNVs, translocations and *IGH* rearrangements.

*SNVs and Indels:* In total, 272 somatic mutations were found across 124 patients (see Appendix A), with an average of 1.7 (0–9) mutations per patient and at least one somatic mutation in approximately 82% of patients. Most of the mutations (253, 93%) were exonic and nonsynonymous SNVs, and 19 (7%) were short indels. As expected, the genes more frequently mutated in terms of percentage of patients carrying at least one mutation were *NRAS* (26%), *KRAS* (23%), *TP53* (12%), *BRAF* (11%), *DIS3* (8%), and *ATM* (8%).

*Translocations and copy number variations:* The NGS panel allows the identification of the three most frequent translocations described in MM, t(4;14), t(11;14) and t(14;16), and CNV identification was mainly focused on regions across chromosome arms 1p, 1q and 17q. Del(1p) was identified in 11 (7%) patients, whereas gain(1q) was observed in 39 (26%) patients (five patients showed both del(1p) and gain(1q)). Del(17p) was observed in eight (5%) patients. Regarding translocations, t(4;14) breakpoints were identified in 12 (8%) patients, t(11;14) breakpoints in 15 (10%) patients, and t(14;16) breakpoints in two (<2%) patients (Appendix A).

*IGH VDJ rearrangements*: Overall, 185 functional *IGH* rearrangements were identified in 121 patients (81%) (Appendix A). The number of rearrangements identified with MiXCR was larger than with Vidjil (165 vs. 117) and likewise was the number of patients (119 vs. 107) in which at least one rearrangement was identified. Rearrangements identified by MiXCR showed a higher average frequency (56% vs. 20%), but this could have been due to the final reads used to assemble the rearrangements (188 vs. 533 on average). A literature-based comparative study of major *IGH* clusters is presented in Appendix A shows a similar clonal detection rate between both techniques. In addition, *IGH* rearrangements observed in some of the patients were validated through PCR-based sequencing.

### 3.3. Comparative Results for NGS Panel, FISH and WES

*Panel versus WES:* Paired tumor-control WES data were available for 51 patients. Within the targeted regions, 47 somatic mutations were identified in the 51 patients using the NGS panel, and 37 of them (79%) were identified by WES (Appendix A). No somatic mutation was found in the WES data that was not present in the NGS panel data. The mean variant allele frequency (VAF) and depth for the NGS panel data were 0.23 (0.01–0.84) and 567× (134–920×), respectively, whereas the VAF and depth for the WES data were 0.26 (0.01–0.88) and 52× (9–182×), respectively. In the case of discordant positions (VAF = 0 in exome data), the mean VAF and depth were 0.04 (0.01–0.14) and 622 (166–891), respectively, for the NGS panel data, whereas in exome data, the average depth was 31 (10–50). The observed frequencies for del(1p), gain(1q) and del(17p) in the exome data were 10%, 29% and 4%, respectively, which were very similar to the frequencies obtained by the targeted approach (8%, 26% and 5%, respectively).

*Panel versus FISH*: Using FISH data as a reference, the NGS panel showed positive (PPV) and negative (NPV) predictive values of 0.90, with a sensitivity of 0.62, a specificity of 0.98 and a global accuracy of 0.90 (Table 2 (A)). For translocations only, PPV, NPV, sensitivity, specificity and global accuracy were 0.92, 0.93, 0.65, 0.99 and 0.92, respectively, and for CNVs were 0.89, 0.89, 0.61, 0.98 and 0.89, respectively. The combination of high-risk alterations (t(4;14), t(14;16), gain(1q), del(17p)) showed very similar concordant values (see Table 2 (A)). When compared to the CNV results from the WES data (Table 2 (B) and Appendix A), PPV, NPV, sensitivity, specificity and global accuracy were 0.97, 0.96, 0.85, 0.99 and 0.96, respectively. Alteration-specific detailed PPV, NPV, sensitivity, specificity and global accuracy are shown in Table 2 and Appendix A.

### 3.4. Impact of Molecular Karyotyping-Identified Alterations on Progression-Free Survival

Survival analyses were performed with data generated by the NGS-targeted panel (SNVs, CNVs and translocations) and with FISH data (CNVs and translocations). Regarding SNVs, only *TP53* mutational status had an impact on PFS (*p* = 0.021, Figure 2A), which was especially significant in cases presenting with homozygous (VAF > 0.5, mostly “double-hit”) mutations in *TP53* (*p* < 0.0001). The impact on PFS of the different structural genomic alterations (t(4;14), t(11;14), t(14;16), del(1p), gain(1q), del(17p)) revealed that only gain(1q) from NGS data was significant (NGS *p* = 0.027 vs. FISH *p* = 0.2, Figure 2A). The impact of del(17p) alone was not significant for PFS independent of the technique (*p* > 0.5, Figure 2A). Using NGS data, we defined a high-risk profile based on classical genetic high-risk alterations (del(17p), gain(1q) and translocations t(4;14)/t(14;16)) adding the mutational status of *TP53*, which significantly predicted (*p* = 0.0015, Figure 2A) the outcome of this cohort of patients. Multivariable analysis (Figure 2B) confirmed the significant impact of the high-risk profile (*p* = 0.015) and of homozygous mutations in *TP53* (*p* = 0.002), multiplying the progression hazard risk by 2.34 (CI = 1.18–4.7, *p* = 0.015) and 6.35 (CI = 1.94–20.9, *p* = 0.002) times, respectively.

## 4. Conclusions

We show here that our targeted NGS panel based on molecular karyotyping can detect, in a single assay and with high precision, the key genomic alterations for MM stratification, including CNVs, SNVs, *IGH* clonal rearrangements and translocations. Moreover, the identification of an MM high-risk genetic profile with the NGS panel showed a suitable correlation with current gold standard techniques: FISH for CNVs and translocations and WES for CNVs and SNVs. Other targeted sequencing panels were previously reported [15,16,34] but with limitations in all cases, such as the inability to detect either translocations, CNVs or *IGH* rearrangements.

The genes often affected in MM are deeply covered by our panel, allowing the detection of both SNVs and indels at a similar frequency to that described in the literature. As expected, the comparative analysis with WES data revealed a greater sensitivity in coding regions targeted by our panel. WES-limited depth across targeted genes could have hampered the detection of mutations, with a VAF of 1–5% [2,35]. According to our results, current standard WES parameters and mean base depth would make it an unreliable tool for the identification of mutations within relevant genes in MM.

The sensitivity shown for translocations reached an average of 60% when compared to FISH data. In this regard, the small number of positive cases (23%) might have negatively affected estimates of the sensitivity to detect translocations. Published results are similar [15,34], pointing out the difficulty in detecting translocations by NGS. Since the nature of the genomic regions prone to produce translocation breakpoint is complex (masked, repetitive homologous regions), the efficiency of the whole process depends on random facts such as where DNA breakpoints occur during the fragmentation step or the efficiency of the probes specific of these regions.

Moreover, we observed no bias in the identification of different events, although del(1p) and t(14;16) presented a lower sensitivity. Taking into account the results obtained from CNVs using WES data, we concluded that this low sensitivity is probably due to FISH limitations in the case of del(1p), derived from the location of the probe; and panel limitations in the case of t(14;16). We are currently working on different improvements in the design of the NGS panel that could increase the sensitivity to detect some of the translocations and CNVs.

On the other hand, we found excellent global accuracy, PPV and NPV (approximately 90%) for the NGS panel when the results were compared with available FISH data. Given that FISH analysis has a limit of detection of approximately 10%, and that FISH is only able to show the presence of a maximum of two concomitant alterations in the same cell clone, we consider that our NGS panel can be a valid diagnostic tool.

The NGS panel was superior to FISH for CNV detection, which was confirmed by WES data. This is likely because FISH probes are designed in very specific locations within chromosomes and partial deletions/gains can occur in more telomeric/centromeric regions that do not affect the FISH signal. An example of this is shown in Appendix A, in which partial deletions/gains were identified by WES and the NGS panel but not by FISH. Additionally, as shown in Appendix A, alterations spreading a shorter length could be identified more precisely with either FISH or NGS panel data that go beyond the resolution of WES. The importance of these specific cases is shown by the difference observed in the impact on PFS. In our cohort of patients, the impact of gain (1q) was significant (*p* = 0.027) using panel data (see Figure 2A) and not when using FISH.

Disease progression is a key event in MM and the stratification of high-risk profiles remains an important goal to guide clinical decision-making [5]. Our results show that the accuracy of the NGS panel for the identification of high-risk patients with short PFS is superior to that of the other genetic techniques. Notably, our approach was able to identify patients with poor outcomes even though this series of patients received highly effective treatments [21].

The main limitation was the small number of true positive cases for some specific alterations, leading to a smaller sensitivity than recent studies [36]. Increasing the density of probes in certain regions for CNV and breakpoint detection is the next step. In return, a strength of our study is the fact that all the patients were homogeneously treated in a clinical trial.

In conclusion, molecular karyotyping by our targeted captured-based NGS panel specific for MM identifies SNVs, CNVs, *IGH* clonal rearrangements and translocations relevant to MM management. Although this approach has the potential to replace conventional cytogenetic approaches for the definition of genomic high-risk genomic profiles, simplifying the genetic study of MM, improvements in panel design will be addressed to improve the sensitivity to detect translocation events.

## Figures and Tables

**Figure 1 cancers-14-05169-f001:**
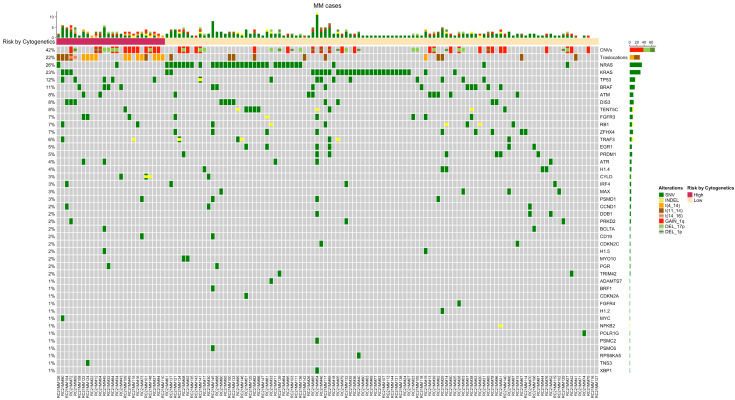
Molecular characterization. Patients were grouped based on the cytogenetic progression risk. CNVs and translocations are shown at the top. Alterations are represented in different colors.

**Figure 2 cancers-14-05169-f002:**
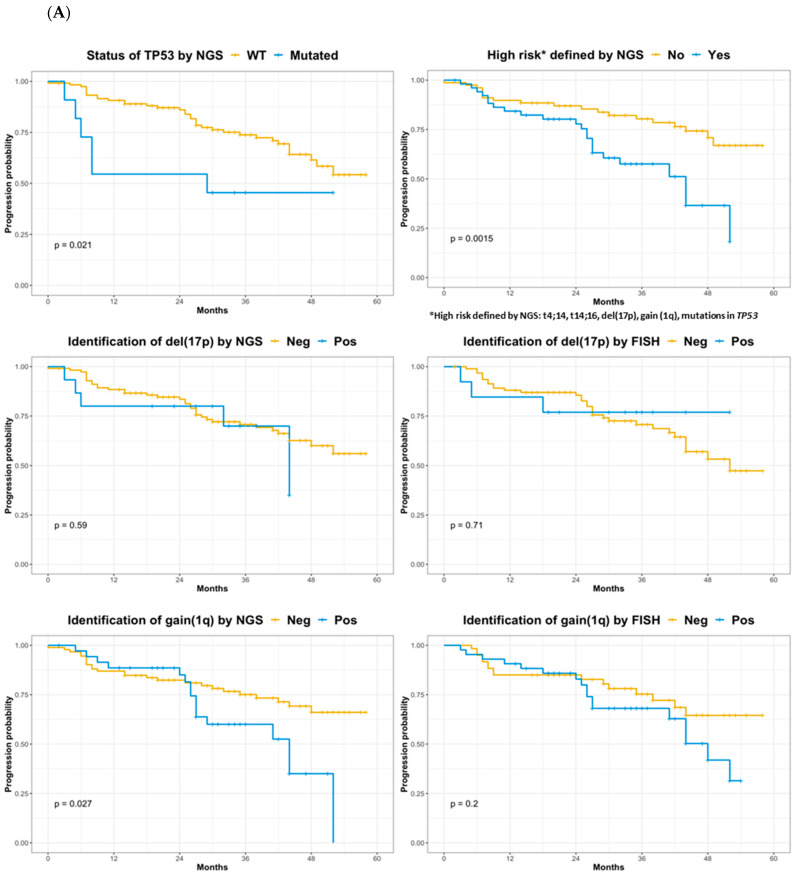
Impact of specific alterations on the progression of MM. (**A**) Univariate survival analysis (Kaplan–Meier plots). *TP53* mutations and high-risk by NGS plots are shown at the top; del(17p) identified by NGS or FISH plots are shown in the middle; and gain(1q) identified by NGS or FISH plots are shown at the bottom. (**B**) Multivariable analysis using *coxph* and *ggforest* functions. Selected values were sex, hemoglobin (high > 9 g/dL), creatinine (high > 1 mg/dL), International Staging System stage (high = ISS III), LDH (lactate dehydrogenase, high = abnormal), *TP53* mutational status, genetic high-risk, and total number of mutations.

**Table 1 cancers-14-05169-t001:** Main clinical patient characteristics. ECOG, Eastern Cooperative Oncology Group; ISS, International Staging System.

	Total in Our Study (N = 149) *	Total GEM2012MENOS65 Clinical Trial (N = 458) *
**Median age (range) in years**	60 (42–65)	58 (31–65)
**Sex (%)**		
Male	52.6	52.4
Female	47.4	47.6
**ECOG performance status (%)**		
0	47	42.6
1	35.6	39.7
2	15.2	13.5
3	2.3	3.5
Missing	2.3	0.7
**M-protein type (%)**		
IgG	63.7	59.6
IgA	20	23.4
Light chain	14.8	15.1
IgD	0	0.7
Nonsecretory	1.5	1.3
**ISS stage (%)**		
I	24.7	23.4
II	36	36.2
III	39.6	39.1
Missing	0	1.3
**Lactate dehydrogenase elevated (%)**		
Yes	8.1	14.2
No	77.8	82.1
Missing	14.1	3.7
**High-risk cytogenetics (%)**	19.7	20.1

* No signficant differences were found between groups.

**Table 2 cancers-14-05169-t002:** Comparative estimators of our NGS panel against FISH and WES. (A) Estimated through comparison to FISH data. (B) Estimated through comparison to exome data. PPV = positive prognostic value, NPV = negative prognostic value. Tx = Translocations.

(A)	**Alteration**	**PPV**	**NPV**	**Sensitivity**	**Specificity**	**Global Accuracy**
	Tx + CNVs	0.90	0.90	0.62	0.98	0.90
	High-risk	0.88	0.91	0.65	0.98	0.91
	Tx	0.92	0.93	0.65	0.99	0.92
	t(4,14)	0.92	0.91	0.61	0.99	0.91
	t(11,14)	0.90	0.85	0.82	0.92	0.87
	t(14,16)	1.00	0.96	0.40	1.00	0.96
(B)	**Alteration**	**PPV**	**NPV**	**Sensitivity**	**Specificity**	**Global Accuracy**
	CNVs	0.97	0.96	0.85	0.99	0.96
	chr1p	1	0.93	0.73	1	0.94
	chr1q	1	0.94	0.88	1	0.96
	chr17	0.83	1	1	0.98	0.98

## Data Availability

The data not presented in this article can be shared up on request.

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
