# Peer review of "NGS-Based Molecular Karyotyping of Multiple Myeloma: Results from the GEM12 Clinical Trial"

_cancers, 2022, doi:10.3390/cancers14205169_

Round 1

Reviewer 1 Report

In the present paper the authors present an analysis of NGS-based karyotyping for newly diagnosed multiple myeloma patients. This is very interesting, as the availability of reliable NGS panels could improve diagnostic of multiple myeloma enabling the simultaneous detection of translocations, CNV and the most relevant mutations. An added strength of the paper is that the population analysed is very homogeneous, having been treated within a clinical trial.

The authors should briefly report which treatment was used in the trial, as most of the readers might not know it. How many patients did also receive a maintenance treatment within the companion trial?

The sensitivity of the methods seems with ca 60% relatively low. The authors should address this aspect more in the discussion. What could be the reasons of the low sensitivity and how could they be overcome? If the idea would be to replace FISH with this method, then is a problem that ca 30% of FISH positive patients are missed with the NGS. I would appreciate if the discussion would focus more on this issue

What are the costs of the panel used? How do they compare with FISH costs in this specific case?

Please check all the references. They are inserted in the paper with a different font and most of the time the surname of the first author is not completely spelled and only the initial of the first name are reported

How was the purity of the sorted plasma cells assessed?

Page 3 line 108-109: I would add “ […] the companion PETHEMA/GEM2014MAIN clinical trials” to make clear that all patients were treated homogenously even to those readers not familiar with the study protocols.

Page 4 line 136-138: “Genes considered to be treatment targets or candidates for drug resistance in MM (PSMD1, XBP1, PSMB5, PSMC2, 137 PSMC6, DDB1, IKZF1 and IKZF3)19,21,22” What do 19, 21 and 22 mean? Do they refer to references?

Page 5 lines 199-213: I assume the number in brackets are the ranges. Please specify.

Page 5 lines 202-203, lines 206-207 and lines 211-212: Please add the range after 2427, 239, 75, 66 and 80.

In the multivariate analysis, how was high and low of the different parameters defined? Was ISS high only ISS III or also ISS II? Please give the reference values what was the cut off for all the different variables

Legends to Supp. Figure 1 and 2 are missing. The figures should probably put on a PDF with the figure legends

Author Response

In the present paper the authors present an analysis of NGS-based karyotyping for newly diagnosed multiple myeloma patients. This is very interesting, as the availability of reliable NGS panels could improve diagnostic of multiple myeloma enabling the simultaneous detection of translocations, CNV and the most relevant mutations. An added strength of the paper is that the population analysed is very homogeneous, having been treated within a clinical trial.

The authors should briefly report which treatment was used in the trial, as most of the readers might not know it. How many patients did also receive a maintenance treatment within the companion trial?

We have added “Summarizing, patients received VRd plus autologous stem cell transplantation, VRd consolidation and maintenance with Lenalidomide or Lenalidomide plus Ixazomib.” To Material and Methods.

The sensitivity of the methods seems with ca 60% relatively low. The authors should address this aspect more in the discussion. What could be the reasons of the low sensitivity and how could they be overcome? If the idea would be to replace FISH with this method, then is a problem that ca 30% of FISH positive patients are missed with the NGS. I would appreciate if the discussion would focus more on this issue

We have modified some of the pragraphs:

“Sensitivity shown for translocations reached an average of 60% when compared to FISH data. In this regard, the small number of positive cases (23%) might have negatively affected estimates of the sensitivity to detect translocations. Published results are similar [18,37], pointing out the difficulty in detecting translocations by NGS. Since the nature of the genomic regions prone to produce translocation breakpoint is complex (masked, repetitive homologous regions), the efficiency of the whole process depends on random facts like where DNA break points occur during the fragmentation step or the efficiency of the probes specific of these regions,

Moreover, we have observed no bias in the identification of different events, although del(1p) and t(14;16) presented a lower sensitivity. Taking into account the results obtained from CNVs using WES data, we concluded that these low sensitivity is probably due to FISH limitations in the case of del(1p), derived from the location of the probe; and panel limitations in the case of t(14;16). We are currently working in different improvements in the design of the NGS panel that could increase the sensitivity to detect some of the translocations and CNVs.

On the hand, we found excellent global accuracy, PPV and NPV (approximately 90%) for the NGS panel when the results were compared with available FISH data. Given that FISH analysis has a limit of detection of approximately 10%, and that FISH is only able to show the presence of a maximum of two concomitant alterations in the same cell clone, we consider that our NGS panel can be a valid diagnostic tool.”

What are the costs of the panel used? How do they compare with FISH costs in this specific case?

In out lab, FISH and NGS costs are very similar (from 300 to 600€ both, depending on number of samples per patient, mainly), but we are not sure if this is something generalizable for each diagnostic laboratory. For that reason, we preferred not to discuss this fact.

Please check all the references. They are inserted in the paper with a different font and most of the time the surname of the first author is not completely spelled and only the initial of the first name are reported

It seems Citation Manager reconfigured citations wrongly. We have changed the style to Cancers.

How was the purity of the sorted plasma cells assessed?

As stated in Material and Methods, “Plasma cells from BM aspirates were enriched using anti-CD138+ immunomagnetic beads, obtaining a purity >85% in all cases.”

Page 3 line 108-109: I would add “ […] the companion PETHEMA/GEM2014MAIN clinical trials” to make clear that all patients were treated homogenously even to those readers not familiar with the study protocols.

We have added it as suggested.

Page 4 line 136-138: “Genes considered to be treatment targets or candidates for drug resistance in MM (PSMD1, XBP1, PSMB5, PSMC2, 137 PSMC6, DDB1, IKZF1 and IKZF3)19,21,22” What do 19, 21 and 22 mean? Do they refer to references?

We apologize, it seems we had problems with the citation manager, and we missed this mistake.

Page 5 lines 199-213: I assume the number in brackets are the ranges. Please specify.

We have specified those were ranges.

Page 5 lines 202-203, lines 206-207 and lines 211-212: Please add the range after 2427, 239, 75, 66 and 80.

We have added all the ranges.

In the multivariate analysis, how was high and low of the different parameters defined? Was ISS high only ISS III or also ISS II?

ISS high included only ISS III

Please give the reference values what was the cut off for all the different variables

We have added this to the Figure 1 legend: “hemoglobin (high > 9 g/dL), creatinine (high > 1 mg/dL), International Staging System stage (high = ISS III), LDH (lactate dehydrogenase, high = abnormal).”

Legends to Supp. Figure 1 and 2 are missing. The figures should probably put on a PDF with the figure legends

Supp. Figure legends were in the manuscript body:

Supp. Figure 1: Distribution of major IGH gene clusters involved in the rearrangements identified in 5 different studies. The results are shown for MiXCR and Vidjil, as both tools were used in our study. No significant differences were observed.

Supp. Figure 2: Visual representation of CNV data in chromosomes 1 and 17 from two different patients. Panel I represents log-ratio values for normalized depth in exome (top) and panel (bottom) data. Panel II represents loss of heterozygosity analysis based on germline polymorphisms (MAF >0.3 and <0.7) from exome data. Red lines are considered lost regions, and green lines are considered gained regions.

We have added in Supp. Figures JPGs

Reviewer 2 Report

In this paper, the authors designed a capture-based NGS targeted panel to identify known genetic alterations for the prognostic stratification of multiple myeloma (MM).  The panel was validated in a cohort of 149 MM patients enrolled in a clinical trial and showed high accuracy results compared to available data from fluorescence in situ hybridization and whole-exome sequencing. Overall, this draft is well written. I think

this panel is useful to identify relevant prognostic chromosomal abnormalities for the clinical management of MM patients.

I only have below minor concerns and they mainly focus on the technical part.

1.

Lines 158-159: “and filtered considering the following parameters: population frequency ≤0.01, total coverage at position ≥50, frequency of the variant in the sample ≥0.01”

Please justify why “population frequency ≤0.01” is used as one of the filtering parameters. As the authors used many population-related databases (such as dbSNP, 1000 Genomes, ESP6500, ExAC, etc.), it is also not clear which population does the authors used to calculate the population frequency.

2.

Line 164: “…potential rearrangements, selecting those events supported by more than 10 reads, …”

Line 167-168: “IGH rearrangements were evaluated using MiXCR(Duez et al, 2016) and Vidjil(Duez et al, 2016), ruling out those supported by <5 reads …”

The read coverage cutoffs seem very arbitrary. The authors applied >=50X for variants detected using VarScan2 and bcftools, >10 reads for potential rearrangements, and >=5 reads for IGH rearrangements.  The authors need to explain why different filtering criteria were applied here.

3.

The methods section “2.5. Whole-exome sequencing” is not clearly described.  The authors mentioned the sequencing yield was ~0.6 Gb per sample but didn’t provide detailed information in terms of the alignment ratio, on-target ratio, mean coverage, etc., at the individual sample level. Therefore, it is hard to make judgement for this section and the comparative results between NGS panel and WES.

In my opinion, the sequencing coverage of the WES data seems very low.  The authors did mention that “Bioinformatics analyses were performed as described above.” However, if WES data went through the same filtering criteria as the NGS panel (such as “total coverage at position ≥50”), I am surprised to see 61 out of 72 somatic mutations (84%) were identified by the WES data.   

4.

The resolution is a bit low for Figure 2.

Author Response

Reviewer 1:

1.

Lines 158-159: “and filtered considering the following parameters: population frequency ≤0.01, total coverage at position ≥50, frequency of the variant in the sample ≥0.01”

Please justify why “population frequency ≤0.01” is used as one of the filtering parameters. As the authors used many population-related databases (such as dbSNP, 1000 Genomes, ESP6500, ExAC, etc.), it is also not clear which population does the authors used to calculate the population frequency.

 We have specified in the text “GMAF population frequency ≤0.01  (obtained from any database)”

2.

Line 164: “…potential rearrangements, selecting those events supported by more than 10 reads, …”

Line 167-168: “IGH rearrangements were evaluated using MiXCR(Duez et al, 2016) and Vidjil(Duez et al, 2016), ruling out those supported by <5 reads …”

The read coverage cutoffs seem very arbitrary. The authors applied >=50X for variants detected using VarScan2 and bcftools, >10 reads for potential rearrangements, and >=5 reads for IGH rearrangements.  The authors need to explain why different filtering criteria were applied here.

We agree with the reviewer that filtering process might seem arbitrary, but, on the contrary, filtering process has been meticulously designed and tested for each alteration type. We observed that, due to technique biases and bioinformatics limitations, the detection of SNVs is far more accurate that the detection of chromosomal rearrangements, while IGH rearrangements required an specific database where all reads are used to aligned to, reducing rearrangements signals. In addition, SNV false positive rate is higher, due to many noise sources, for that reason, filtering strictness had to be different, and specific, for SNVs, chromosomal rearrangements and IGH rearrangements.

3.

The methods section “2.5. Whole-exome sequencing” is not clearly described.  The authors mentioned the sequencing yield was ~0.6 Gb per sample but didn’t provide detailed information in terms of the alignment ratio, on-target ratio, mean coverage, etc., at the individual sample level. Therefore, it is hard to make judgement for this section and the comparative results between NGS panel and WES.

We have added all the information in the Supp. Table 2.

In my opinion, the sequencing coverage of the WES data seems very low.  The authors did mention that “Bioinformatics analyses were performed as described above.” However, if WES data went through the same filtering criteria as the NGS panel (such as “total coverage at position ≥50”), I am surprised to see 61 out of 72 somatic mutations (84%) were identified by the WES data.  

 We have changed the sentence in Material and Methods to “Bioinformatics analyses were performed as described above, except for total coverage at position that we used ≥10 instead of ≥ 50”

4.

The resolution is a bit low for Figure 2.

We have increased the resolution of Figure 2 to 400dpi.

Reviewer 3 Report

In this manuscript Rosa-Rosa et al. expose their studies on the design of a NGS single assay to identify known genetic alterations for the targeted molecular karyotyping of Multiple Myeloma (MM) patients and for their prognostic stratification. The NGS panel could simultaneously analyze single nucleotide and copy number variations, insertions and deletions, chromosomal translocations and V(D)J rearrangements. They validated the panel in a cohort of 149 MM patients enrolled in the GEM2012MENOS65 clinical trial and demonstrated a great global accuracy and the ability of identifying with the panel high-risk patients that had a shorter progression-free survival.

In my opinion, this study is well-written and even if the NGS panel has some technical limits, as the authors themselves recognize, it lays the foundations for expanding the use of capture-based NGS targeted panels for clinical management of MM patients.

Major comments:

·       11 of 72 somatic mutations were identified using the NGS panel but not by WES. I think that their presence should be validated with other molecular techniques.

·         Figure 1 and Figure 2 resolutions should be improved.

Other minor comments are noted in the pdf file.

Author Response

Major comments:

  •      11 of 72 somatic mutations were identified using the NGS panel but not by WES. I think that their presence should be validated with other molecular techniques.

We are trying to validate some of the mutations in sameples in which we still have remanent DNA at diagnosis (unfortunately not many). In addition, this validation cannot being performed usin regular PCR and sanger sequencing, since the VAF is below sanger sequencing sentitivity threshold. We are currently designing specific primers to detect translocations, IG rearrangements and low frequency SNVs using NGS technology. Although results seem promising, we still have room to improve and are not ready to be publish.

  • Figure 1 and Figure 2 resolutions should be improved.

Resolutions have been improved to 400 dpi.

Minor changes have been applied to manuscript file.

Round 2

Reviewer 3 Report

Where are Figures 1 and 2???? Have you improved their resolution?

Lines 314-315: You have not named Supp. Figure 2 with A and B letters. 

Author Response

- Where are Figures 1 and 2???? Have you improved their resolution?

We submitted both figures in a zip with updated Supp. Tables. Resolution was 400dpi. We submit again.

- Lines 314-315: You have not named Supp. Figure 2 with A and B letters.

We changed the text to (left) and (right)